# Effects of Marginal Bone Loss Progression on Stress Distribution in Different Implant–Abutment Connections and Abutment Materials: A 3D Finite Element Analysis Study

**DOI:** 10.3390/ma15175866

**Published:** 2022-08-25

**Authors:** Ching-Ping Lin, Yi-Ting Shyu, Yu-Ling Wu, Ming-Hsu Tsai, Hung-Shyong Chen, Aaron Yu-Jen Wu

**Affiliations:** 1Department of Dentistry, Kaohsiung Chang Gung Memorial Hospital and Chang Gung University College of Medicine, Kaohsiung 833, Taiwan; 2Department of Mechanical Engineering, Cheng Shiu University, Kaohsiung 833, Taiwan; 3Center for Environmental Toxin and Emerging-Contaminant Research, Cheng Shiu University, Kaohsiung 833, Taiwan

**Keywords:** stress distribution, bone loss, implant–abutment connection, zirconia abutment

## Abstract

Peri-implantitis is a common implant-supported prosthesis complication, and marginal bone loss affects the stress distribution in implant systems. This three-dimensional finite element analysis study investigated how bone loss affects the implant assembly; in particular, models including two implant systems with different connection systems (external or internal hexagon), abutment materials (titanium or zirconia), and bone loss levels (0, 1.5, 3, or 5 mm) were created. We observed that the maximum von Mises stress distinctly increased in the groups with bone loss over 1.5 mm compared to the group without bone loss, regardless of the connection system or abutment material used. Moreover, the screw stress patterns with bone loss progression were determined more by the connection systems than by the abutment materials, and the magnitude of the stress on the fixture was affected by the connection systems with a similar pattern. The highest stress on the screw with the external hexagon connection system increased over 25% when bone loss increased from 3 to 5 mm, exceeding the yield strength of the titanium alloy (Ti–6Al–4V) when 5 mm bone loss exists; clinically, this situation may result in screw loosening or fracture. The highest stress on the fixture, exceeding the yield strength of pure titanium, was noted with the internal hexagon connection system and 1.5 mm bone loss. Titanium and zirconia abutments—both of which are clinically durable—presented similar screw and fixture stress patterns. Therefore, clinicians should pay more attention to maintaining the peri-implant bone to achieve the long-term stability of the implant-supported prosthesis.

## 1. Introduction

For congenitally missing teeth or loss of teeth due to reasons such as trauma, caries, and periodontal diseases, the restorative options include crowns and bridges, dental implants, removable partial dentures, autotransplantation, and space closure via orthodontic treatment. The popularity of dental implants has increased over the last few decades because they aid in regaining the function and aesthetics of a single missing tooth or a partially edentulous area, both without affecting neighboring teeth [1].

An implant-supported single crown comprises a crown, abutment, screw, and fixture. Both increasing the survival rate and decreasing the complication rate of implant-supported single crowns are clinically crucial. The 10-year survival rates of implant-supported single crowns and fixed dental prostheses range from 89% to 94% [2]. The criteria for the success of osseointegrated endosseous implants include the following: upper limit of the bone resorption during the first year of loading <1.5 mm, mean annual vertical bone loss after the first year <0.2 mm, and implant design providing satisfactory prosthesis [3,4]. Implant-supported prosthesis complications can be classified into biological, technical, and aesthetic [4]. Notably, loosening of the abutment and screw is a technical complication with the highest incidence [5,6].

An ideal implant material should be biocompatible with adequate toughness, strength, and resistance to the corrosion and fracture [7]. Titanium is nontoxic and more biocompatible than chromium-cobalt and stainless steel in the dental industry. Titanium implants have dominated dental implantology since the 1960s. Titanium remains the gold standard for implant fabrication, and the zirconia implants may be a promising alternative in the future [8]. A wide range of abutment materials is currently available; they include titanium, gold, alumina, and ceramic. Ti–6Al–4V is an alpha-beta titanium alloy with a high strength-to-weight ratio and favorable corrosion resistance. It is the most-used titanium alloy of implant abutments and screws. Furthermore, ceramic is more fragile than titanium; as such, ceramic abutments lead to significantly higher risks of abutment fractures than do metal abutments [2]. Because metal abutments may compromise the aesthetic outcome for some patients, ceramic abutments such as those composed of zirconia are preferred by most patients [9].

The implant–abutment connection is critical for implant-supported fixed prosthesis outcomes. An abutment may be connected to the implant in an internal or external system. Internally connected abutments have relatively more biomechanical advantages, such as a lower incidence of screw loosening [10]. In the case of a single crown, both metal and zirconia abutments have been noted to demonstrate high survival rates with internal or external connections [2].

Peri-implantitis, an unwanted biological dental implant complication, is defined as “a pathological condition occurring in tissues around dental implants, characterized by inflammation in the peri-implant connective tissue and progressive loss of supporting bone” [11]. Bone loss extending over 3 mm apical to the implant platform is considered to indicate a peri-implant disease [11,12]. Peri-implantitis incidence ranges from 0.4% at 3 years to 43.9% at 5 years, and its prevalence ranges from 1.1% at 3 years to 85.0% at 5 years [13]; peri-implantitis shows an increasing tendency with implant-supported prosthesis functioning [14]. Bone loss results in a decreased contact surface of the implant and surrounding bone, and the mechanical loading affects the stress distribution at the implant–bone interface and in the implant system. A previous three-dimensional (3D) finite element analysis (FEA) study reported that in pure vertical bone resorption models, stress over both cortical and cancellous bone significantly increased with bone resorption progression and the stress increased under lateral loading as bone resorption advanced [15]. Bone resorption can alter the implant–crown ratio and significantly affect the biomechanical behavior of the implant-supported prosthesis, potentially causing implant failure [16]. Studies have investigated the effect of peri-implant bone resorption on the stress distribution in the implant system by using FEA methods [16,17,18,19]. The stress transition from the implant to the surrounding bone has been noted to be key in long-term stability. However, the patterns of the effects of peri-implant bone loss on the stress distribution over the implant assembly with different implant designs such as implant–abutment connection systems and component materials remain unclear.

The current study, therefore, evaluated the biomechanical behavior including the stress distribution and magnitude in the implant assembly by using 3D FEA with different abutment materials (titanium or zirconia), abutment–implant connection systems (internal or external hexagon), and horizontal bone loss magnitude (0, 1.5, 3, or 5 mm).

## 2. Materials and Methods

### 2.1. 3D Model Design

The implant components were measured under a digital microscope (VHX-900F; Keyence Corporation, Osaka, Japan) and vernier calipers (Mitutoyo 505-752; Mitutoyo Corporation, Kanagawa, Japan) and scanned using a 3D optical scanning system (Aicon SmartScan-HE; Breuckmann, Germany) to obtain high-resolution images; these images were used to construct a 3D FEA model by using computer-aided design software (inventor2020; Autodesk, San Rafael, CA, USA) with FEA (ANSYS Workbench 2020 R1; ANSYS, USA).

Two implant systems with different implant–abutment connections were used (Figure 1): NobelSpeedy Groovy RP 4 mm × 10 mm with external hexagon (S group) and NobelParallel Conical Connection RP 4.3 mm × 10 mm with internal hexagon (C group; both from Nobel Biocare, Kloten, Switzerland). We also used two abutment materials: Snappy titanium abutments (Snappy abutments; Nobel Biocare, Yorba Linda, CA, USA; T group) and zirconia abutments (simulated using Young’s modulus and Poisson’s ratio of zirconia in previous studies; Z group). The screw (Nobel Biocare, Kloten, Switzerland) connecting the abutment and implant fixture was composed of Ti–6Al–4V. A full zirconia crown was fixed over the abutment to resemble the morphology of a maxillary central incisor.

The implant fixture was placed in the bone housing model of 25 mm (height) × 12 mm (width) × 10 mm (thickness) with the cancellous bone surrounded by a 2 mm cortical bone at four levels from the implant platform to simulate different amounts of horizontal bone loss: 0 mm (no bone loss), 1.5 mm, 3 mm, and 5 mm (Figure 2 and Figure 3). In total, 16 models in four groups were designed in this study, and these models are denoted according to their connection system–abutment material–bone loss magnitude (e.g., “ST0” for NobelSpeedy Groovy-external hexagon–titanium–0 mm bone loss; Table 1).

### 2.2. Material Properties

All materials used here were set to be isotropic, homogenous, and linearly elastic in this FEA study. The implant fixture, abutment, screw, and crown materials were verified through energy-dispersive X-ray spectroscopy (JSM-6360; JEOL, Tokyo, Japan). Although yielding is clearly defined for metals, plastic, and ceramic materials, it is questionable for live tissues such as the heterogeneous bone structure [20]. The mechanical properties used in this FEA study were extracted from previous literature (Table 2).

### 2.3. Elements and Nodes

The meshes of all the components were constructed using tetrahedron elements (SOLID187). The tetrahedron is a high-order 3D, 10-node element with a quadratic displacement behavior and is well suited to modeling irregular meshes.

The ST and SZ group comprised a similar number of elements, as did the CT and CZ groups (Table 3).

### 2.4. Interface Conditions

The implant–abutment, implant–screw, and abutment–screw interfaces were set as contacts with a coefficient of friction of 0.3 [25]. The implant fixture with ideal osseointegration in the bone model was assumed, and the implant–bone interface was set as a contact. For the implant surface–cortical bone and implant surface–trabecular bone interfaces, the coefficients of friction were set at 0.65 [26] and 0.77 [27], respectively.

### 2.5. Loading and Boundary Conditions

According to the suggestions of the manufacturer, the screw tightening torque was applied at a torque of 0.35 N·m. The axial force was calculated using the following equation [28]: T = KDF, where T is the tightening torque (N·m), K is the torque coefficient, D is the screw diameter (m), and F is the axial force (N). Axial force on the S-group and C-group screws was set at 499.8 N and 492.2 N, respectively. A 170 N lateral force was applied on the palatal surface of the crown at a 45° oblique angle to the long axis of the implant (Figure 4).

### 2.6. FEA

In all 16 models (listed in Table 1), we applied the same loading conditions, boundary conditions, and constraints. The elastic modulus was set to be linear because the structure did not have a large deformation or plastic failure under loading. The von Mises stress in all models was measured over all the implant assembly components. ANSYS Workbench 2020 R1 (ANSYS, Inc., Canonsburg, PA, USA) was used to analyze the data and perform the stress analysis.

## 3. Results

### 3.1. Overall Stress Distribution Pattern of Each Implant Assembly Component with Bone Loss Progression

The magnitudes of the von Mises stress on each component of implant assembly and surrounding cortical and cancellous bone under identical loading conditions are presented in Table 4 and Figure 5, Figure 6 and Figure 7.

We observed that the connection system determined the changes in the screw stress distribution pattern with bone loss progression. In the ST and SZ groups, the increase in stress was gradual as bone loss increased from 0 to 3 mm, but it became sharp as bone loss increased from 3 to 5 mm. By contrast, in the CT and CZ groups, stress increased as bone loss increased from 0 to 1.5 mm, but it decreased as bone loss increased from 1.5 to 5 mm.

With regard to stress on the fixture with bone loss progression, the patterns were similar in all four groups: the stress significantly increased as bone loss increased from 0 to 1.5 mm and then at a similar level as bone loss increased from 1.5 to 5 mm. Nevertheless, the von Mises stress magnitudes were all larger in the CT and CZ groups than in the ST and SZ groups—indicating that stress on the fixture was higher with an internal hexagon connection system than with an external hexagon connection system.

With regard to stress on the abutment with bone loss progression, the patterns remained similar in all the groups, except the CT group, which demonstrated a constant increase in stress with bone loss progression. In general, the von Mises stress magnitude increased with bone loss progression, reaching its maximum value at 3 mm of bone loss, but decreasing with an increase in bone loss from 3 to 5 mm. The patterns were most similar between the SZ and CZ groups.

### 3.2. Stress Distribution Pattern of Each ST Model Component with Different Bone Loss Levels

In the ST models, the highest von Mises stress occurred on the screw, followed by that on the abutment or fixture for all bone loss levels (Figure 8). The locations of the maximum von Mises stress on the abutment, screw, and fixture are displayed in Figure 9.

The location of the maximum stress on the screw moved apically with bone loss, from the implant head–neck connection region with no (0 mm) bone loss to the shank–thread connection region with any amount of bone loss. The magnitude of the stress increased stably with bone loss progression and sharply increased by 31.6%, from model ST3 (657.49 MPa) to model ST5 (864.76 MPa).

The location of the maximum stress on the fixture remained unchanged with bone loss progression: coronal border of the first tread of the fixture. The magnitude of the stress increased significantly from model ST0 (315.35 MPa) to model ST1.5 (561.82 MPa); it remained constant even with further bone loss.

The location of the maximum stress on the abutment remained almost unchanged with bone loss progression: the screw head–abutment contact region. The magnitude of the stress was 38.5% higher in model ST3 (646.06 MPa) than in model ST1.5 (466.38 MPa); however, this magnitude decreased to 625.07 MPa in model ST5. In other words, the highest stress on the abutment was noted in model ST3.

Finally, the ST3 model led to the highest stress on the bone.

### 3.3. Stress Distribution Pattern of Each CT Model Component with Different Bone Loss Levels

In the CT models, the highest and lowest von Mises stress occurred on the fixture and abutment for all bone loss levels, respectively (Figure 10). The locations of the maximum von Mises stress on the abutment, screw, and fixture are displayed in Figure 11.

The location of the maximum stress on the screw remained unchanged with bone loss progression: the neck–thread junction. The magnitude of the stress increased with bone loss from 0 to 3 mm, and the stress in model CT3 (759.67 MPa) was higher than that in model CT5 (725.37 MPa).

The location of the maximum stress on the fixture also remained unchanged with bone loss progression: the fixture’s buccal coronal inner border. The magnitude of the stress increased significantly from model CT0 (623.89 MPa) to CT1.5 (856.26 MPa); it remained constant in models CT3 (855.76 MPa) and CT5 (854.95 MPa).

The location of the maximum stress on the abutment also remained unchanged with bone loss progression: contact position of the coronal border of the screw head and the abutment thread. The stress magnitude increased consistently with bone loss progression.

Finally, the model CT5 led to the highest stress on the bone.

### 3.4. Stress Distribution Pattern of Each SZ Model Component with Different Bone Loss Levels

In the SZ models, the highest von Mises stress occurred on the screw, followed by that on the abutment or fixture for all bone loss levels (Figure 12). The patterns of the stress distribution and locations of maximum von Mises with regard to bone loss progression in the SZ models were almost identical to those in the ST models (Figure 13).

The stress on the screw increased stably as bone loss increased from 0 to 3 mm; it demonstrated a considerable increase from 676 MPa to 858.02 MPa from 3 to 5 mm.

The stress on the fixture first increased significantly from 316.39 MPa in model SZ0 to 576.54 MPa in model SZ1.5, then decreased to 521.97 MPa in model SZ3, and finally, increased slightly to 546.26 MPa in model SZ5.

The stress on the abutment in models SZ0 and SZ1.5 was almost identical; it gradually increased to 581.15 MPa in model SZ3 and decreased to 549.94 MPa in model SZ4. The highest stress on the abutment was noted in model SZ3.

Finally, the SZ3 model exhibited the highest stress on the bone.

### 3.5. Stress Distribution Pattern of Each Component of the CZ Model with Different Bone Loss Levels

In the CZ models, the highest von Mises stress occurred on the screw, followed by that on the fixture and, finally, by that on the abutment (Figure 14). The locations of maximum von Mises stress with bone loss progression in the CZ models were almost identical to those in the SZ models (Figure 15).

The location of the maximum stress on the screw remained unchanged with bone loss progression: neck–thread junction. The magnitude of the stress increased considerably from 593.91 MPa in model CZ0 to 758.36 MPa in model CZ1.5, but decreased to 727.79 MPa in model CZ5.

The stress on the fixture significantly increased by 34.9%, from 590.51 MPa in model CZ0 to 796.64 MPa model CZ1.5; this value remained almost unchanged regardless of further bone loss.

The stress on the abutment gradually increased from 508.63 MPa in model CZ0 to 612.64 MPa in model CZ3, but decreased slightly to 554.16 MPa in model CZ5.

Finally, the CZ5 model exhibited the highest stress on the bone.

## 4. Discussion

FEA can be used to simulate the loading conditions over implant components and the surrounding bone in different modified conditions; it allows for the manipulation of the different parameters on designed models [29,30]. Understanding the stress distribution in implants and the peri-implant bone can aid in improving implant design and provide useful information for clinical consideration. The 16 FEA models created in this study included implant assembly with two implant connection system types, two abutment material types, and four bone loss levels. We used these models to evaluate the stress distribution in the implant system three-dimensionally and compared how the three factors affected the von Mises stress on the each of implant components.

Among the available implant–abutment systems, external hexagon implants demonstrated many advantages such as simplicity, predictability, anti-rotation mechanism, and compatibility with different systems. However, they are also associated with a 6–48% complication rate, with the complications including conditions such as screw loosening [31]. In the 1990s, Mollersten et al. indicated the strength advantage of internal connections [32]. The internal connection design enables a loading distribution deep into the implant assembly; this leads to the shielded screw condition such that the load on the screw becomes lower than that when an external connection is used. In their FEA study, Yamanishi et al. indicated that the implant neck design and abutment connection types affect the stress over the bone and abutment [33]. Therefore, we used NobelSpeedy Groovy RP with an external hexagon (ST and SZ groups) and NobelParallel Conical Connection RP with an internal hexagon (CT and CZ groups) to evaluate the effects of the connection system on the stress distribution with different bone loss levels and abutment materials.

In the present study, with the same loading condition, the maximum stress was noted at the palatal side of the screw and at the buccal side of the fixture, regardless of the connection type. The highest maximum von Mises stress was on the screw rather than on the fixture or abutment with the external hexagon, whereas the highest stress was on the fixture rather than on the screw or abutment with the internal hexagon. The increase in the stress on the screw when bone loss increased from 0 to 3 mm was larger in the groups with the external hexagon than with the internal hexagon. By contrast, the stress on the fixture for all bone loss levels was lower in the groups with the external hexagon than with the internal hexagon.

The connection type, therefore, has a larger impact on the distribution of stress on the screw and fixture with different bone loss levels than the abutment material. We noted that as bone loss progressed, stress moved apically in the fixture in the internal hexagon connection system, whereas it remained unchanged on the screw in the external hexagon connection system. However, the impacts of the connection systems on the stress distribution on the abutments remained inconclusive in this study. Studies have indicated that implant abutments in an internal hexagon connection system demonstrate a widely spread force distribution down to the implant compared with those in an external hexagon connection system [34] and that internal connections are more favorable than external connections [35]. Although a recent review article indicated no significant differences un the survival and biological complications among the connection types [36], their impact on the risk of biological complications, mechanical complications (e.g., marginal bone loss), or both should be considered [37].

In recent years, the clinical application of ceramic materials has increased because of the superior aesthetic outcome they afford. Of all available ceramic materials, zirconia-based ceramic exhibits excellent mechanical properties and biocompatibility, and it can be used as a strong implant abutment. The strength and toughness of zirconia are attributable to mechanisms such as crack deflection, contact shielding, zone shielding, and crack bridging [38]. Zirconia abutments are available for the various types of implants, such as external and internal hexagon implants [39]. Although a review indicated that both externally and internally connected ceramic abutments are associated with higher fracture rates than are externally and internally connected metal abutments [2], no statistically or clinically significant differences between survival rates and the incidence of complications of zirconia and titanium abutments have been noted [40]. Nevertheless, the impacts of the zirconia abutments on the implant assembly remain unclear. The oblique forces to an implant may affect the abutment–implant interface. A study suggested that a zirconia abutment–titanium fixture–external hexagon combination causes permanent deformation on the hexagon and causes stress to accumulate over the top of the abutment, potentially resulting in microfractures or microgaps within the abutment [41]. Although the strength of the zirconia abutment may decrease after cyclic loading, it may have sufficient mechanical strength clinically [42].

The yttrium-stabilized tetragonal zirconia polycrystal (Y-TZP) has attracted the attention of a considerable number of clinicians recently because of its strength. Here, we used the elastic modulus and Poisson’s ratio of Y-TZP to simulate our zirconia abutment and to compare its effects with those of our titanium abutment. The patterns of the stress distribution of the screw and fixture with zirconia and titanium abutments were noted to be similar in all models (Figure 5 and Figure 6). This suggested that the screw and fixture are more affected by the connection system than by the abutment material, and the stress on the fixture was generally smaller in the CZ group than in the CT group. Moreover, in both connection systems, the patterns of stress on the zirconia abutment were more similar than the patterns of stress on titanium abutments (Figure 7). The impact of the zirconia abutment material might be higher than that of the connection system with bone loss progression. In contrast, the impact of abutment material on the stress pattern was not obvious for the titanium abutment. In general, stress on the zirconia abutment in all models was much lower than the flexure strength of the Y-TZP (900–1200 MPa [43], 1120 MPa (Nobel Biocare)), whereas that on the titanium abutments was lower than the yield strength of the titanium alloy Ti-6Al-4V (795 MPa). Thus, the application of both titanium and zirconia abutments, which are easily available and are safe for clinical use, and their impact on the screw and fixture are nonsignificant. However, wear may occur at the one-piece zirconia abutment–titanium fixture interface [44]. Periodic recalls for examination of the implant assembly are recommended.

As the period of the implant in function increases, so do the complication risks. A prevalent complication is peri-implant bone loss. Constituents of the implant system are key in the implant’s and the surrounding bone’s stability. Because the connection system affects the stress distribution, the impact of bone loss with regard to the external and internal hexagon connections warrants further discussion.

The location of the maximum stress on the abutment and fixture remained unchanged with different bone loss levels, but that on the screw moved from the implant head–neck connection to the shank–thread connection (Figure 9 and Figure 13). The maximum stress with the external hexagon was observed on the screw at all bone loss levels (Figure 8 and Figure 12). In the ST groups, the maximum rate of change in the stress on the abutment was noted when the bone loss increased from 1.5 to 3 mm (38.5% increase) and from 3 to 5 mm (31.5% increase), whereas the maximum rate of change in the stress on the fixture was noted when bone loss increased from 0 to 1.5 mm (78.2% increase). Similar patterns were noted in the screw and fixture in the SZ groups, whereas the stress patterns of the abutment demonstrated generally gradual increases in the SZ group compared with that in the ST groups. Notably, the stress on the screw increased with bone loss progression; it increased sharply when bone loss increased from 3 to 5 mm, and the magnitude of the stress on the screw with 5 mm bone loss exceeded the yield strength of Ti-6Al-4V (795 MPa). A bone loss of >5 mm may therefore cause catastrophic damage to the screw, such as screw fracture; in such cases, surgical removal or replacement of the screw may be required [45].

In the internal hexagon systems (in the CT and CZ groups), the locations of the maximum stress on the abutment, screw, and fixture remained unchanged with bone loss progression (Figure 11 and Figure 15); the only exception was the location of the maximum stress on the abutment in the CZ models, which shifted to a more apical site with bone loss progression (Figure 15). For all bone loss levels, the stress on the screw and abutment was lower than that on the fixture. The maximum stress on the screw of CT and CZ groups was observed with a 3 mm bone loss. However, the stress location on the fixture did not change with bone loss progression, but its magnitude exceeded the yield strength of pure titanium (680 MPa) when bone loss exceeded 1.5 mm. Notably, here, the stress location was the border of the fixture head, which is the thinnest part of a fixture [46]. Higher stress at the vertex may cause a higher tendency to crack, developing microfractures and even microgaps over the implant-abutment connection, which may create a window for bacterial penetration [47]. Therefore, when using an internal hexagon connection system, clinicians should consider the impact of bone loss on the fixture because the resulting stress might cause damage over time.

Most of previous investigations focused on the stress in the peri-implant bone with different bone levels, and there were few studies investigating the stress magnitude and distribution pattern of the component of the implant assembly with different bone levels [16,17,18,19,48,49]. Our result that stress increases with bone loss progression in the components of the implant corroborates that of previous studies. Yenigun et al. reported that the stress magnitude of the implant body, screw, and abutment increased with bone loss progression, and the maximum stress location of the abutment was at the interface between the first thread of the screw and the abutment shank. Moreover, they suggested that stress can jeopardize implant assembly integrity when the bone loss exceeds 3 mm [17]. Bing et al. indicated that as peri-implant bone loss extends, the stress in the implant body and screw increased [48]. In the present study, regardless of the abutment material used, the stress caused damage to the fixture when bone loss exceeded 1.5 mm in an internal hexagon connection system, and it caused damage to the screw when bone loss exceeded 5 mm in an external hexagon connection system. Michilidis et al. indicated that with >2 mm peri-implant bone resorption, both periodontal tissue and implant prosthesis led to critical stress, which might lead to high implant fatigue and possible fracture, even under mild loading [16]. The stress on the bone and implant system increases as bone loss progresses, and the position moves on the bone level circumferences with the exposed fixture thread. The risks of implant fracture can particularly increase under lateral loading [49]. Gupta et al. indicated that implant–bone assemblies might not survive in the long term if bone loss exceeds 25% of the implant length [18]. In conclusion, the impact of bone loss, which is reflected by the peri-implant bone condition, is an essential factor affecting the stress distribution in an implant assembly.

The limitations of the present study are as follows: (1) An FEA study design has inherent limitations related to the simplifications and assumptions used therein. The parameters in the FEA study were set to be absolute (e.g., a perfectly bonded bone–implant interface) without any consideration of different clinical conditions (e.g., overload due to parafunction). Although this study focused on the stress on the implant assembly, the bone design can be improved in the future. Pietroń et al. recently addressed a method with tomography and algorithms, which can be used in the improvement of determining the parameters of the bone model design. The parameters of the heterogeneous bone structure may be determined according to different individuals or regions [50]. In other words, our FEA study could not reflect the current clinical scenarios completely. (2) There are multiple different bone loss patterns observed clinically. However, simplified bone loss patterns with total horizontal bone loss were chosen in this study. Although pure horizontal bone loss leads to low stability, the shape of the bony defect might influence the stress distribution near the neck of the implant [15]. Additional studies considering different patterns of bone loss are thus warranted. (3) We applied the Y-TZP parameters reported previously to the zirconia abutment. Many types of zirconia abutments are available, and several factors may influence their stability and survival; for instance, a recent study indicated that the fracture resistance of a zirconia abutment with a titanium insert is higher than that of an only-zirconia abutment [51]. Some investigations indicated that zirconia is a promising alternative material to titanium used as the implant, but without long-term clinical trials [52]. Considering the current daily clinical situation, zirconia was only used as the abutment material in our study. Hence, future studies should consider different zirconia abutment designs and zirconia implants.

## 5. Conclusions

Despite the limitations inherent to the FEA study design, the following conclusions were drawn:With marginal bone loss exceeding 1.5 mm, the maximum von Mises stress obviously increases on the screw and fixture regardless of the connection system or abutment materials. Among the factors, peri-implant bone loss affects the magnitude and distribution of the stress on the implant assembly the most.With bone loss progression, the connection system drives the distribution pattern of the stress on the screw. The stress concentrates the least on the abutment among the three components of the implant assembly. Moreover, both titanium and zirconia abutments are safe for clinical use.In this study, the stress on the screw in the external hexagon connection system sharply increased to >25% when bone loss increased from 3 to 5 mm—exceeding the yield strength of titanium alloy (Ti–6Al–4V) and, thus, increasing the screw loosening or fracture risk. Moreover, the stress on the fixture with the internal hexagon connection system sharply increased when the bone loss was ≥1.5 mm—exceeding the yield strength of pure titanium. Therefore, marginal bone maintenance is critical for conserving an implant assembly’s integrity.

## Figures and Tables

**Figure 1 materials-15-05866-f001:**
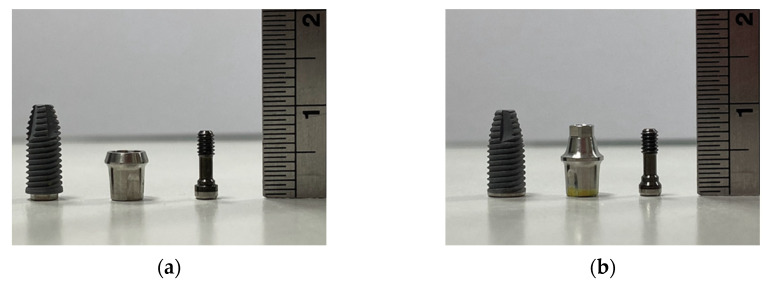
Components of the implant–abutment–screw systems: (**a**) S group: NobelSpeedy Groovy RP 4 mm × 10 mm with external hexagon; (**b**) C group: NobelParallel Conical Connection RP 4.3 mm × 10 mm with internal hexagon.

**Figure 2 materials-15-05866-f002:**
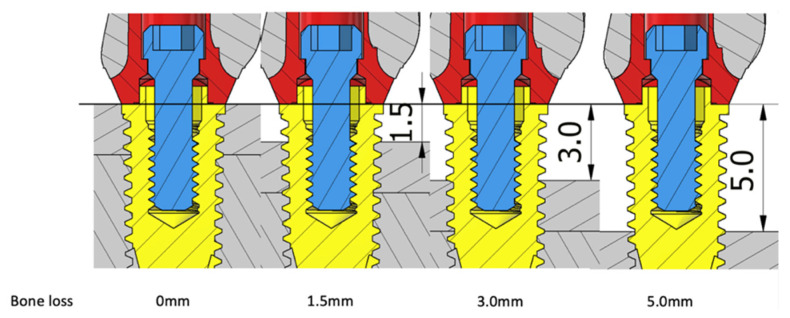
S groups with 0 mm (ST0 and SZ0), 1.5 mm (ST1.5 and SZ1.5), 3 mm (ST3 and SZ3), and 5 mm (ST5 and SZ5) bone loss.

**Figure 3 materials-15-05866-f003:**
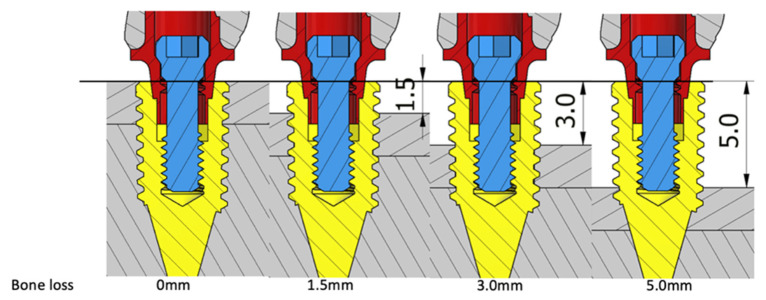
C groups with 0 mm (CT0 and CZ0), 1.5 mm (CT1.5 and CZ1.5), 3 mm (CT3 and CZ3), and 5 mm (CT5 and CZ5) bone loss.

**Figure 4 materials-15-05866-f004:**
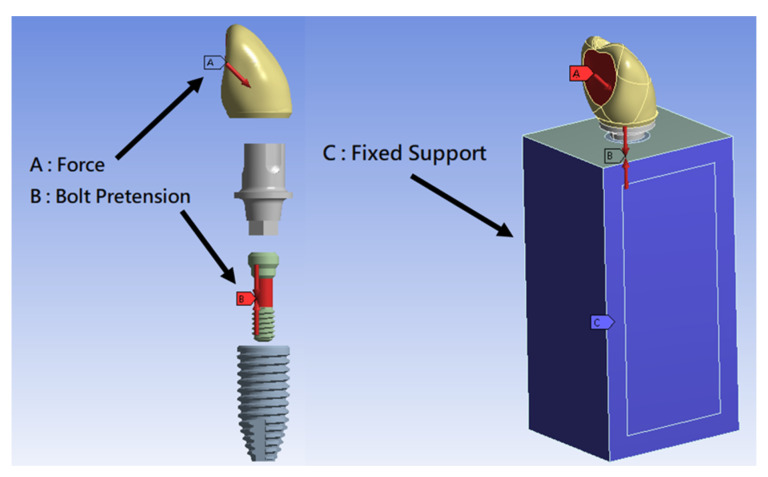
Loading condition on all models (A, lateral force of 170 N applied at the palatal side of the upper incisor crown with 45° to the long axis of the implant; B, bolt pretension with the axial force on the screw; C, fixed supported bone housing model).

**Figure 5 materials-15-05866-f005:**
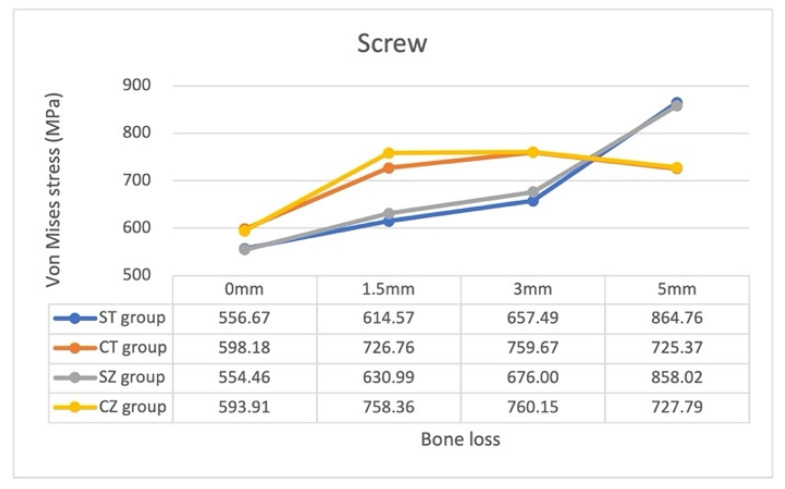
Maximum von Mises stress on the screw of the 16 models.

**Figure 6 materials-15-05866-f006:**
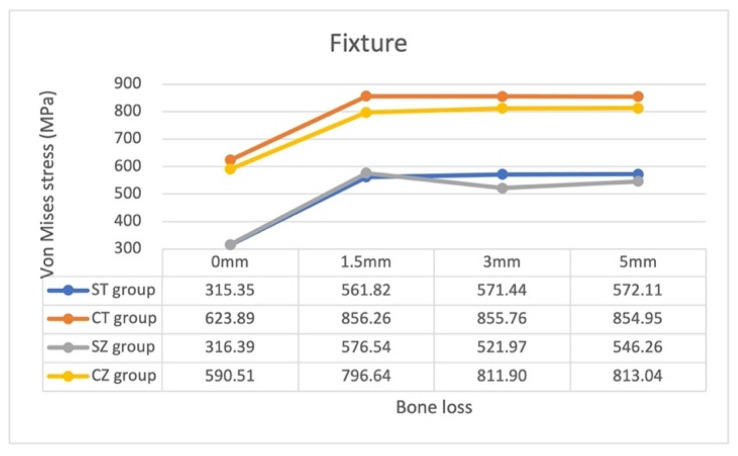
Maximum von Mises stress on the fixture in the 16 models.

**Figure 7 materials-15-05866-f007:**
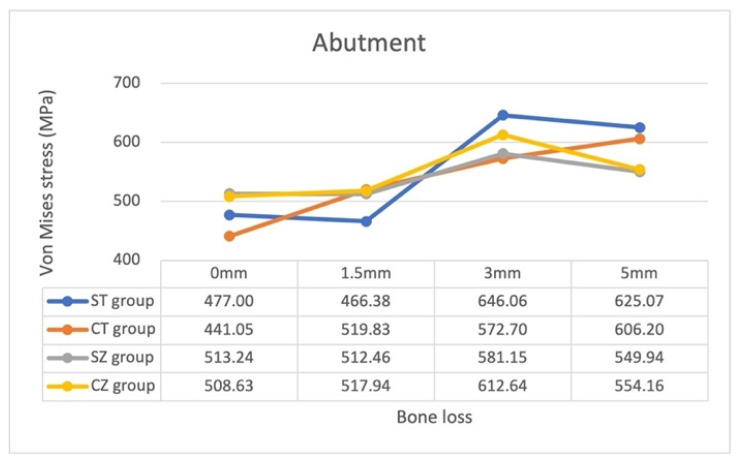
Maximum von Mises stress on the abutment in the 16 models.

**Figure 8 materials-15-05866-f008:**
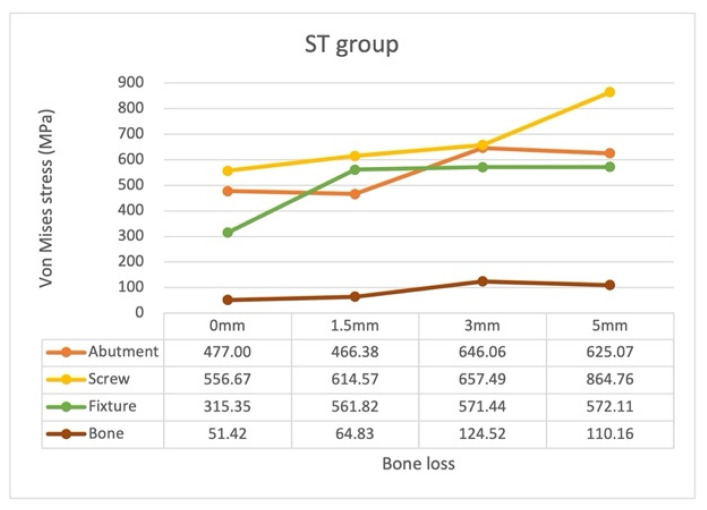
Maximum von Mises stress on the abutment, screw, fixture, and bone in the ST groups with different bone loss levels.

**Figure 9 materials-15-05866-f009:**
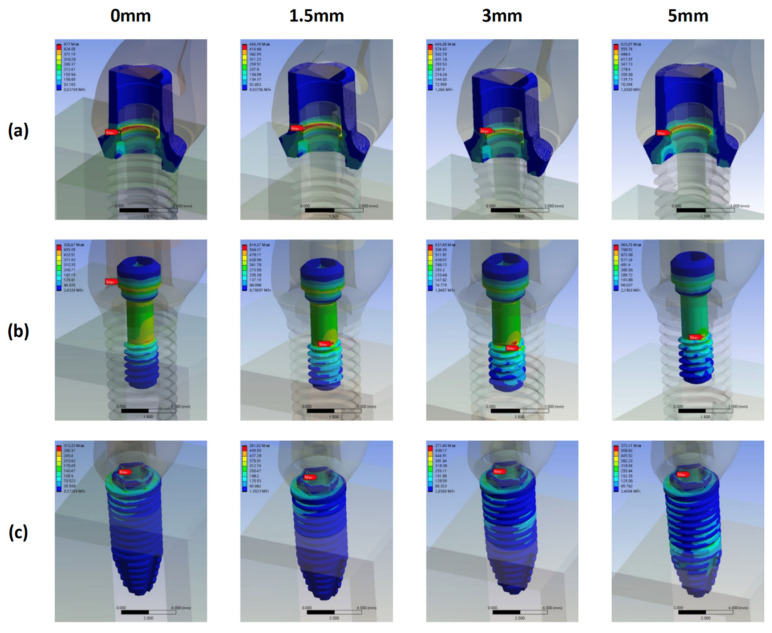
Stress distribution and maximum von Mises stress on the abutment, screw, and fixture in the ST groups with different bone loss levels. (**a**) abutment (**b**) screw (**c**) fixture.

**Figure 10 materials-15-05866-f010:**
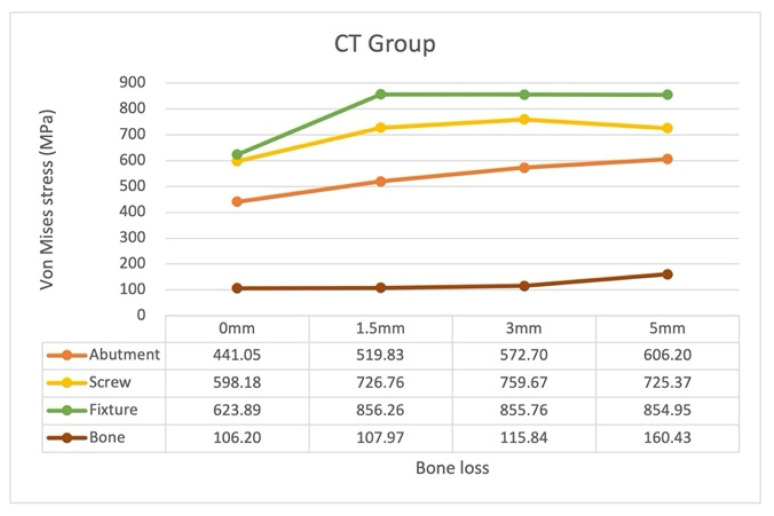
Maximum von Mises stress on the abutment, screw, fixture, and bone in the CT groups with different bone loss levels.

**Figure 11 materials-15-05866-f011:**
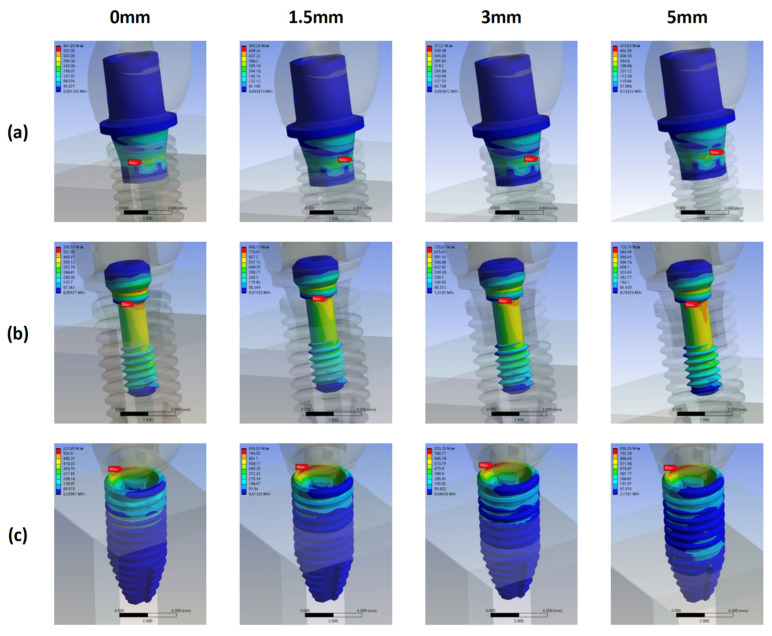
Stress distribution and maximum von Mises stress on the abutment, screw, and fixture in the CT groups with different bone loss levels. (**a**) abutment (**b**) screw (**c**) fixture.

**Figure 12 materials-15-05866-f012:**
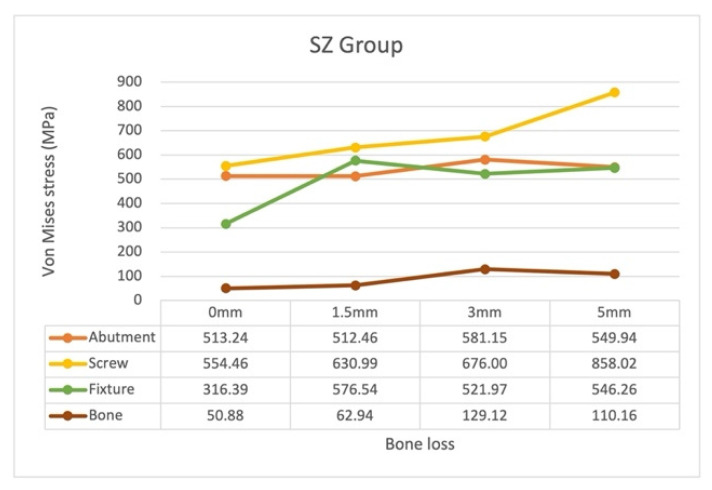
Maximum von Mises stress on the abutment, screw, fixture, and bone in the SZ groups with different bone loss levels.

**Figure 13 materials-15-05866-f013:**
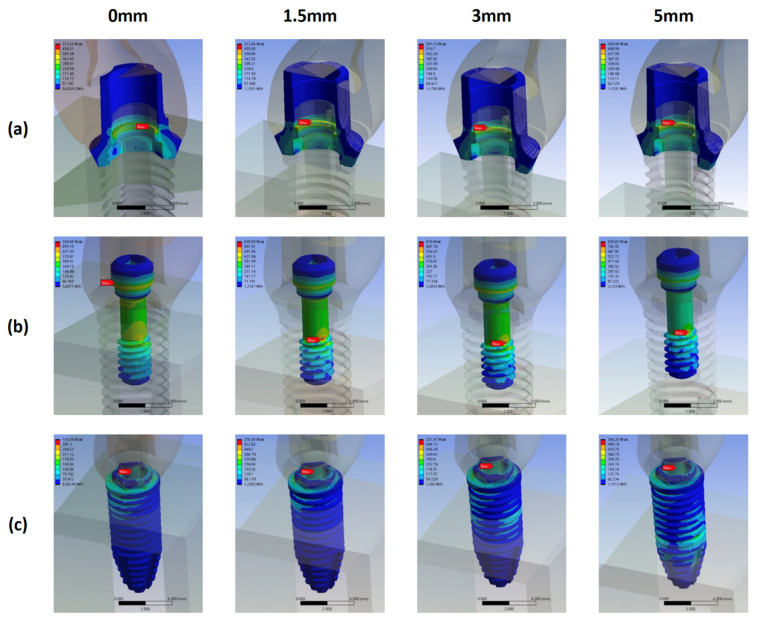
Stress distribution and maximum von Mises stress on the abutments, screws, and fixtures in the SZ groups with different bone loss levels. (**a**) abutment (**b**) screw (**c**) fixture.

**Figure 14 materials-15-05866-f014:**
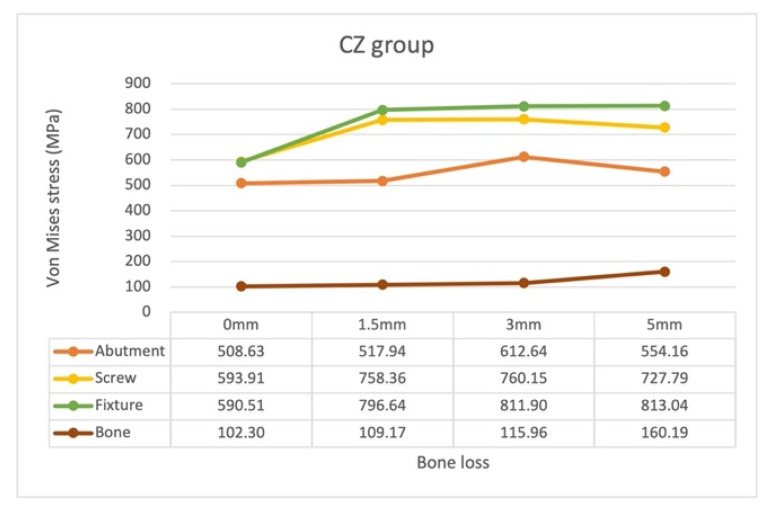
Maximum von Mises stress on the abutment, screw, fixture, and bone in the CZ groups with different bone loss levels.

**Figure 15 materials-15-05866-f015:**
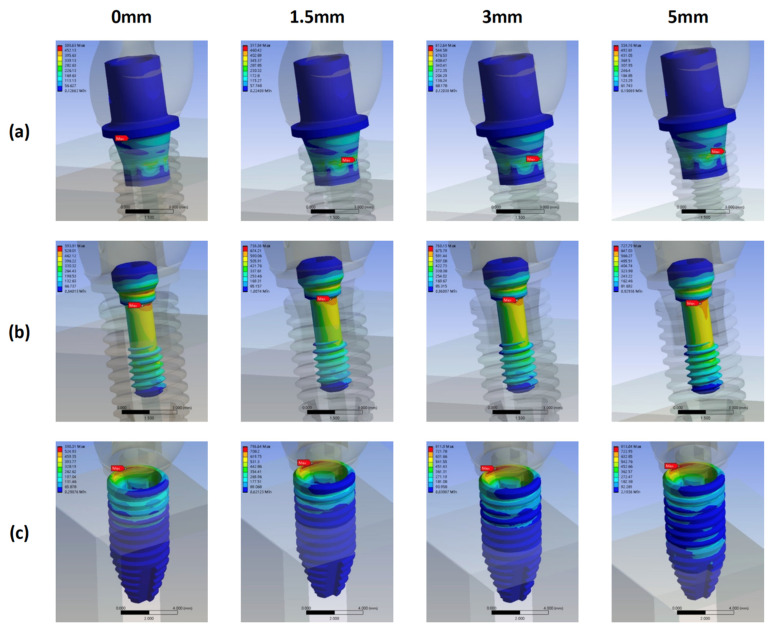
Stress distribution and maximum von Mises stress on the abutments, screws, and fixtures in the CZ groups with different bone loss levels. (**a**) abutment (**b**) screw (**c**) fixture.

**Table 1 materials-15-05866-t001:** Characteristics of the 16 models in the 4 groups.

Group	Implant-Connection	Abutment	Bone Loss
ST0	NobelSpeedy Groovy-external hexagon	Titanium	0 mm
ST1.5	NobelSpeedy Groovy-external hexagon	Titanium	1.5 mm
ST3	NobelSpeedy Groovy-external hexagon	Titanium	3 mm
ST5	NobelSpeedy Groovy-external hexagon	Titanium	5 mm
SZ0	NobelSpeedy Groovy-external hexagon	Zirconia	0 mm
SZ1.5	NobelSpeedy Groovy-external hexagon	Zirconia	1.5 mm
SZ3	NobelSpeedy Groovy-external hexagon	Zirconia	3 mm
SZ5	NobelSpeedy Groovy-external hexagon	Zirconia	5 mm
CT0	NobelParallel Conical Connection-internal hexagon	Titanium	0 mm
CT1.5	NobelParallel Conical Connection-internal hexagon	Titanium	1.5 mm
CT3	NobelParallel Conical Connection-internal hexagon	Titanium	3 mm
CT5	NobelParallel Conical Connection-internal hexagon	Titanium	5 mm
CZ0	NobelParallel Conical Connection-internal hexagon	Zirconia	0 mm
CZ1.5	NobelParallel Conical Connection-internal hexagon	Zirconia	1.5 mm
CZ3	NobelParallel Conical Connection-internal hexagon	Zirconia	3 mm
CZ5	NobelParallel Conical Connection-internal hexagon	Zirconia	5 mm

**Table 2 materials-15-05866-t002:** Materials used in this study.

Material	Young’s Modulus (GPa)	Poisson’s Ratio	Yield Strength (MPa)	Ultimate Strength (MPa)
Cortical bone	13.4 [21]	0.30 [21]	N/A	121; 167 [21] *
Cancellous bone	1.37 [21]	0.30 [21]	N/A	N/A
Pure Titanium (implant fixture)	115 [22]	0.35 [22]	Min.750 **	Min.860 **
Ti-6Al-4V alloy (screw, abutment)	110 [23]	0.33 [23]	Min.795 **	Min.860 **
Zirconia (abutment)	200 [24]	0.31 [24]	N/A	1120 ***

* The ultimate tensile and compressive strength of cortical bone (121 MPa and 167 MPa, respectively) [21]. **According to the manufacturer (Nobel Biocare). ***According to the manufacturer, the flexural strength (biaxial) of yttria-stabilized tetragonal zirconia polycrystal (YTZP): 1120 MPa.

**Table 3 materials-15-05866-t003:** Element numbers of finite element models.

Group	Elements (Sum)	Group	Elements (Sum)
ST0	209,327	CT0	232,736
ST1.5	213,471	CT1.5	236,901
ST3	195,462	CT3	233,716
ST5	191,566	CT5	232,922
SZ0	210,982	CZ0	232,736
SZ1.5	203,038	CZ1.5	237,038
SZ3	193,602	CZ3	233,716
SZ5	191,621	CZ5	232,922

**Table 4 materials-15-05866-t004:** Maximum von Mises stress (MPa) on the components of the 16 models.

Group	Abutment	Screw	Fixture	Cortical Bone	Cancellous Bone
ST0	477.00	556.67	315.35	51.42	1.85
ST1.5	466.38	614.57	561.82	64.83	4.00
ST3	646.06	657.49	571.44	124.52	8.27
ST5	625.07	864.76	572.11	110.16	13.99
SZ0	513.24	554.46	316.39	50.88	1.88
SZ1.5	512.46	630.99	576.54	62.94	4.82
SZ3	581.15	676.00	521.97	129.12	7.79
SZ5	549.94	858.02	546.26	110.16	13.99
CT0	441.05	598.18	623.89	106.20	2.51
CT1.5	519.83	726.76	856.26	107.97	4.61
CT3	572.70	759.67	855.76	115.84	8.18
CT5	606.20	725.37	854.95	160.43	16.72
CZ0	508.63	593.91	590.51	102.30	2.57
CZ1.5	517.94	758.36	796.64	109.17	4.60
CZ3	612.64	760.15	811.90	115.96	8.18
CZ5	554.16	727.79	813.04	160.19	16.74

## Data Availability

Correspondence and requests for materials should be addressed to Aaron Yu-Jen Wu.

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
