# Peer review of "Effects of Marginal Bone Loss Progression on Stress Distribution in Different Implant–Abutment Connections and Abutment Materials: A 3D Finite Element Analysis Study"

_materials, 2022, doi:10.3390/ma15175866_

Round 1

Reviewer 1 Report

This is a very well written manuscript demonstrating a simulation method of marginal bone loss and stress distribution in different Implant–Abutment connections and abutment materials.  Authors should emphasize the strict simulation nature of this experiment and several assumptions that were made to run this simulation.  I am also not sure what software and version were used. 

Reviewer 2 Report

1.     Figure 2 – all numbers should have the same number of digits after the decimal point.

2.     Table 2 – this table should include also yield stress and ultimate stress to better understand content of the article, and results from table 4.

3.     The authors used tetrahedron elements. Add information about type of used elements.

4.     Torque is given in unit N·cm. Please use the SI unit of N·m.

5.     What was the value of K (torque coefficient)?

6.     Figure 4 is unreadable. Maybe “picture in picture” in the image will help?

7.     The results in table 4 for the bone should be given separately for cancellous and cortical bone.  

8.     Figures 5-8, 10, 12, 14 – remove zeros after the decimal point.

9.     Figures 9, 11, 13, 15 are too small and illegible.

10.                 Set all materials as isotropic, homogenous, and linearly elastic is oversimplification. This is especially true in the case of bone material. In general, bone is not isotropic. Bones are made of bone tissue that is formed during development and growth. They adapt to the loads they carry, so their structures are quite varied. However, under specific conditions, for bones of the same density, the isotropic and homogenous behavior can be assumed (DOI: 10.3390/ma15155163).

Linear elastic behavior can also be assumed if the stresses do not exceed the limit of proportion. Please clarify that in your research no material exceeds this limit.

Reviewer 3 Report

The authors have presented an interesting study where they have evaluated the effect of abutment material on stress values and areas using a FEA model. Some comments/questions below for the authors:

1) How were the loading and boundary conditions determined for the study? Was effect of cycles or cyclic load considered for the study?

2) Why was only the effects of different abutment material considered and not different materials for screw and implant considered?

3) How were the different bone loss amounts determined?

4) The authors mention in their concluding remarks that wit bone loss progression, maximum Von Mises Stress increased on the assembly. However, this statement is not accurate as the maximum stress increased only in some cases and pretty much remained steady/similar in most of the cases. 

5) I understand there are certain limitations with FEA models but have the authors compared the general trend for failures in the dental implant system and do they corelate with the findings of this study? 

Reviewer 4 Report

In general, it has been a quality and high level work. I have given details at several points. Please make minor revision.

Round 2

Reviewer 2 Report

In previous version the torque was 35 [N cm]. Now it is 3500 [Nm].

I think that it is a mistake. It should be 350 [N mm] or 0.35 [Nm].
